# Photochemical inactivation as an alternative method to produce a whole-cell vaccine for uropathogenic *Escherichia coli* (UPEC)

Marlena M. Westcott,[1] Alexis E. Morse,[1] Gavin Troy,[1] Maria Blevins,[2] Thomas Wierzba,[2] John W. Sanders[2]

**ABSTRACT**   Uropathogenic *Escherichia coli* (UPEC) is the primary causative agent of lower urinary tract infection (UTI). UTI presents a serious health risk and has considerable secondary implications including economic burden, recurring episodes, and overuse of antibiotics. A safe and effective vaccine would address this widespread health problem and emerging antibiotic resistance. Killed, whole-cell vaccines have shown limited efficacy to prevent recurrent UTI in human trials. We explored photochemical inactivation with psoralen drugs and UVA light (PUVA), which crosslinks nucleic acid, as an alternative to protein-damaging methods of inactivation to improve whole-cell UTI vaccines. Exposure of UPEC to the psoralen drug AMT and UVA light resulted in a killed but metabolically active (KBMA) state, as reported previously for other PUVA-inactivated bacteria. The immunogenicity of PUVA-UPEC as compared to formalin-inactivated UPEC was compared in mice. Both generated high UPEC-specific serum IgG titers after intramuscular delivery. However, using functional adherence as a measure of surface protein integrity, we found differences in the properties of PUVA- and formalin-inactivated UPEC. Adhesion mediated by Type-1 and P-fimbriae was severely compromised by formalin but was unaffected by PUVA, indicating that PUVA preserved the functional conformation of fimbrial proteins, which are targets of protective immune responses. *In vitro* assays indicated that although they retained metabolic activity, PUVA-UPEC lost virulence properties that could negatively impact vaccine safety. Our results imply the potential for PUVA to improve killed, whole-cell UTI vaccines by generating bacteria that more closely resemble their live, infectious counterparts relative to vaccines generated with protein-damaging methods.

**IMPORTANCE**   Lower urinary tract infection (UTI), caused primarily by uropathogenic *Escherichia coli*, represents a significant health burden, accounting for 7 million primary care and 1 million emergency room visits annually in the United States. Women and the elderly are especially susceptible and recurrent infection (rUTI) is common in those populations. Lower UTI can lead to life-threatening systemic infection. UTI burden is manifested by healthcare dollars spent (1.5 billion annually), quality of life impact, and resistant strains emerging from antibiotic overuse. A safe and effective vaccine to prevent rUTI would address a substantial healthcare issue. Vaccines comprised of inactivated uropathogenic bacteria have yielded encouraging results in clinical trials but improvements that enhance vaccine performance are needed. To that end, we focused on inactivation methodology and provided data to support photochemical inactivation, which targets nucleic acid, as a promising alternative to conventional protein-damaging inactivation methods to improve whole-cell UTI vaccines.

**KEYWORDS**   UPEC, uropathogenic *E. coli*, urinary tract infection, vaccine, psoralen, AMT, UV light, photochemical inactivation, formalin

Address correspondence to Marlena M. Westcott, mwestcot@wakehealth.edu.

The authors declare no conflict of interest.

See the funding table on p. 16.

Lower UTI is one of the most common infections for which individuals seek medical care (1). Uropathogenic *Escherichia coli* (UPEC) accounts for approximately 80% of lower UTI cases of bacterial origin (2). Clinically uncomplicated lower UTI manifests as cystitis (infection of the urinary bladder) with positive urine culture and localized genitourinary symptoms. Lower UTI can lead to life-threatening infection of the upper urinary tract (pyelonephritis) and urosepsis (2). Greater than 50% of women suffer from UTI in their lifetime, often with recurrent episodes (rUTI) that substantially impact quality of life (2, 3). UTI is a significant health issue for the elderly due to multiple risk factors including loss of bladder control, hormonal changes to urogenital tissue, poor hygiene, decreased immune function, and comorbidities such as diabetes mellitus (4). Antibiotics negatively impact normal vaginal and gastrointestinal flora, and treatment for recurrent infections or treatment for bacteriuria without symptoms has promoted the emergence of resistant strains (5). A safe and effective vaccine to prevent rUTI targeted to women of all ages is needed to address a substantial health and economic burden (3, 6) and to help slow the emergence of antibiotic-resistant strains.

Mouse models of UPEC-induced acute and chronic bladder infection (7, 8) have aided in the development of vaccine candidates based on virulence proteins that support a life cycle with extracellular and intracellular phases (9, 10). For example, Type I fimbriae (T1F) are required for adhesion to and invasion of bladder epithelial cells (11), and antibodies raised against the FimH tip adhesion are protective in mice and nonhuman primates (12, 13). Surface receptors and secreted siderophores required for iron acquisition are protective when administered to mice (14–16). Other potential targets of the pathogenic cycle include an α-hemolysin that promotes the infection and persistence of UPEC within bladder epithelial cells (17, 18) and factors that promote the formation of intracellular and extracellular biofilm structures (19) that protect bacteria from innate immune cells and antibiotics (20, 21). Information from mouse models, which recapitulate aspects of the human host response to UTI (22, 23), have led to advances such as the addition of adjuvants that skew cellular immunity toward protective Th1-type responses combined with delivering vaccines by mucosal routes for optimal activation of local immune responses in the bladder (15, 16, 24). However, due in part to the complex UPEC infectious cycle involving multiple virulence proteins and their heterogeneity across clinical strains (10, 25), advancing UTI vaccine candidates based on individual proteins toward clinical translation has remained a challenge.

In contrast, multiple whole-cell-based UTI vaccine candidates have moved to human clinical trials in the past several decades [reviewed in reference (26)]. Most have been tested in women with a history of rUTI and consist of either bacterial cells killed with protein-targeting methods or bacterial cell lysates (UPEC alone or mixed with other uropathogenic bacteria). Prattley et al., conducted a meta-analysis of the results of 17 clinical trials with four different whole-cell or lysate-based UTI vaccines and concluded that the vaccines were safe and reduced the risk of recurrent infection over the short term (<6 months) (26). For example, MV140/Uromune, consisting of heat-killed UPEC, *Klebsiella pneumoniae*, *Enterococcus faecalis*, and *Proteus vulgaris* self-administered daily for 3 months by the sublingual route reduced the risk of rUTI in women, including elderly, as compared to antibiotic prophylaxis or placebo controls [reviewed in refrence (27)]. OM-89 (UroVaxom) is an oral capsule consisting of a mixture of membrane proteins prepared from lysates of 18 UPEC strains that is taken daily for 90 days followed by several booster doses. At 12 months, a 34% reduction in infections versus placebo was observed in women with a history of rUTI (28, 29). Solco-Urovac consists of 10 heat-killed uropathogens (6 UPEC strains + *K. pneumonia*, *P. mirabilis*, *M. morganii*, and *E. faecalis*) delivered by vaginal suppository in three weekly doses, followed by 3 monthly boosters. At 6 months, 46% of women who received the vaccine remained UTI-free as compared to 17% of the placebo group (30). Collectively, the results with whole-cell-based UTI vaccines are encouraging; however, none of the vaccines tested in human trials to date are licensed for use in the United States indicating an opportunity to improve the whole-cell platform.

Given positive safety and efficacy data from human trials, and the advantage of whole cell over subunit vaccines with respect to presenting a full repertoire of protein and polysaccharide antigens to the immune system, we explored the possibility of improving the platform by addressing the inactivation method. Conventional inactivation with formalin or heat negatively impacts vaccine immunogenicity by modifying or destroying protein antigenic epitopes (31–36). We hypothesized that photochemical inactivation, which targets nucleic acid rather than protein, could improve whole-cell UTI vaccines by sparing proteins from modification. Treatment with psoralens, a class of small molecule, membrane-permeable drugs, followed by irradiation with long wavelength UVA light (365 nm), introduces crosslinks in nucleic acid at pyrimidine residues, disabling pathogen replication while leaving proteins largely intact (37–40). Multiple species of psoralen and UVA light (PUVA)-inactivated bacteria have been tested for their potential as vaccines and have shown efficacy in animal models (41). PUVA-inactivated bacteria have been characterized as killed but metabolically active (KBMA), reflecting sustained gene expression in genomic areas between crosslinks, with *de novo* protein production proceeding hours after inactivation as measured by metabolic labeling and cell-associated dehydrogenase activity (37, 41). Some studies have shown that the KBMA property is consequential for vaccine performance. For example, KBMA *Bacillus anthracis* gave rise to an immune response against PA, a secreted protein target of the protective immune response (42). The KBMA property was observed in a study in which *E. coli* was photochemically inactivated with the psoralen drug 8-MOP for the purpose of UTI vaccine development, but no immunogenicity data were reported (43). We recently reported that enterotoxigenic *E. coli* (ETEC) inactivated with the psoralen drug AMT and UVA light were KBMA, retained surface adhesins in native-like form, and demonstrated superior immunogenicity in mice relative to formalin-killed ETEC (44). Here, we have extended the approach to UPEC. We report on the properties and immunogenicity of psoralen-inactivated UPEC as compared to formalin-inactivated UPEC. The results suggest the potential for PUVA to improve UTI vaccine performance by producing bacteria with properties that more closely resemble their live, infectious counterparts.

## MATERIALS AND METHODS

### Bacteria

UPEC CFT073 (O6:H1:K2, ATCC 700928) (45) was used for these studies. Bacteria were cultured in LB Broth at 37°C with shaking unless otherwise noted.

### Preparation of PUVA-inactivated UPEC vaccine

UPEC from an overnight shaking LB culture was diluted 1:500 into fresh LB broth. After 5 h of shaking at 37°C, AMT psoralen (4′-Aminomethyltrioxsalen hydrochloride, Sigma-Aldrich, A4330) was added to early stationary phase bacteria ($OD_{600}$ 1.5) at a final concentration of 50 µg/mL. After 1 h of incubation, bacteria were transferred to individual wells of a six-well tissue culture dish (1.0 mL/well, with lid on to maintain sterility) and irradiated with UVA light (365 nm) at a dose of 2 $J/cm^2$ using a UVA crosslinker (model CL-1000L, Analytik Jena, US). Bacteria were harvested from wells and kept cold for further processing: after washing three times with PBS to remove AMT, the vaccine preparation was resuspended in LB + 25% glycerol and stored in working aliquots at −80°C. Samples were plated undiluted on LB agar to confirm that no residual CFU remained. For AMT dose-response experiments, live and inactivated samples were serially diluted and plated on LB agar to quantitate CFU/mL remaining. Controls included heat-killed UPEC (1 h 65°C), live UPEC (no PUVA treatment), UPEC treated with AMT only, and UPEC treated with UVA light only. For assays using UPEC cultured under static conditions, AMT was added for the final hour of static culture with gentle mixing at 15 min intervals and inactivation proceeded as above.

## Preparation of formalin-inactivated UPEC vaccine

UPEC from an overnight shaking LB culture was diluted 1:500 into fresh LB broth. After 5 h of shaking at 37°C, early stationary phase bacteria ($OD_{600}$ 1.5) were washed 3× with sterile PBS and resuspended to a concentration of $1 \times 10^{10}$ /mL in PBS + 1.5% formalin. After an additional 2 h at 37°C with shaking, bacteria were incubated static at 4°C for 3 days and then washed to remove formalin. The vaccine preparation was resuspended in LB + 25% glycerol and stored in working aliquots at −80°C. Undiluted samples were plated on LB agar to confirm inactivation. Plating on LB agar was performed to quantitate CFU/mL remaining in dose-response experiments.

## Metabolic activity assay for KBMA property

MTS assays for cellular dehydrogenase enzyme activity as a measure of live cell metabolism were performed according to the manufacturer's instructions (Promega CellTiter 96 AQueous nonradioactive Cell Proliferation Assay). The assay was performed in a flat-bottom 96-well plate with triplicate samples. Inactivated or live bacteria were washed and resuspended to $OD_{600} = 2.0$ in LB broth. Wells were seeded with 75 µL of LB Broth, 25 µL of bacteria and 10 µL of MTS reagent. Control wells contained LB broth and MTS reagent. The plate was incubated at 37°C for 2 h, then the absorbance at 490 nm (to detect soluble purple formazan product of dehydrogenase enzyme activity) and 690 nm (reference wavelength) was measured using a BioTek Epoch 2 microplate spectrophotometer (Agilent Technologies, Winooski, VT, USA). The reference wavelength value was subtracted from the $OD_{490}$ value for each well. The average $OD_{490}$ of media control wells (LB + MTS reagent) was subtracted from the reference-corrected $OD_{490}$ value of each test well. The results were plotted as the mean of triplicate $OD_{490}$ values ± S.D. for each sample.

## Immunogenicity in mice

Immunogenicity of PUVA and formalin-killed UPEC vaccines was measured in 6- to 8-week-old female BALB/c mice (Charles River) under Wake Forest School of Medicine ACUC protocol A19-163. Vaccines were thawed on ice, washed three times with cold PBS and adjusted by $OD_{600}$ reading to @ $2 \times 10^8$ CFU equivalents/mL. Mice were primed and then boosted 3 weeks later by the intramuscular (IM) route with 0.1 mL of vaccine consisting of $2 \times 10^7$ CFU equivalents of PUVA-UPEC or formalin-UPEC. Prior to immunization and 2 weeks after the boost, blood was collected, processed to obtain serum, and frozen individually and in pools at −80°C.

## UPEC-specific ELISA

An ELISA assay using whole UPEC as the coating antigen was adapted from a published protocol (46). UPEC cultured in LB broth to early stationary phase (as per the inactivation protocol) were harvested at 5 h, washed, and adjusted to an $OD_{600} = 0.5$ in sterile PBS. Microtiter plates (Maxisorp, NUNC) were coated with the bacterial suspension (100 µL per well) and the plates were tightly sealed to prevent drying. After 18–24 h at 37°C, the wells were washed three times with PBS and blocked with 200 µL/well of 5% Difco skim milk in ELISA wash buffer (PBS + 0.05% Tween 20) for 2 h at 23°C. After three washes, the serum that had been serially diluted in PBS + 0.05% Tween 20 + 1% skim milk (antibody buffer) was added (100 µL/well) and incubated for 2 h at 23°C. The wells were washed three times and incubated with HRP-conjugated goat anti-mouse IgG (Invitrogen) (100 µL/well) diluted 1:5,000 in antibody buffer. After 1 h, the wells were washed and 100 µL of TMB substrate (3,3', 5.5'-tetramethylbenzidine, Sigma, T0440) was added. Plates were incubated in the dark for 15 min and the reaction was stopped by adding 100 µL of a 2 M $H_2SO_4$. Absorption at 450 nm was measured using a BioTek Epoch 2 microplate spectrophotometer (Agilent Technologies, Winooski, VT). The serum antibody titer, defined as the dilution that yielded an $OD_{450}$ value fourfold greater than the assay background (antigen-coated wells with antibody buffer only added), was calculated by

linear regression analysis of OD readings from 3 to 4 Log10-transformed dilutions using GraphPad Prism software. The dilution series for post-vaccination sera was started at 1:500, which generated no background signal in uncoated (no antigen) control wells. Rabbit anti-FimA antiserum (gift from S. Subashchandrabose, Texas A&M) was paired with HRP-conjugated donkey-anti-rabbit IgG (GE Healthcare, NA934OV) to measure FimA as an indication of Type I fimbriae production by UPEC cultured as described.

## Yeast agglutination assay

Assay for agglutination of UPEC with *Saccharomyces cerevisiae* (12, 47) was performed as a measure of mannose-dependent adhesion mediated by UPEC T1F. *S. cerevisiae* strain YSC2 was prepared by suspending colonies from SDA agar in PBS to $OD_{600}$ = 8.0. UPEC was grown in static LB broth to enhance T1F production (47). Bacteria from 48 h static LB cultures (18–24 h, 37°C) were diluted into fresh LB (1:100), cultured for an additional 18–24 h and then adjusted to $OD_{600}$ = 1.5. The agglutination assay was performed in a round bottom 96-well plate with control wells consisting of yeast only and bacteria only. Equal volumes (50 µL) of yeast and bacteria (live, PUVA-inactivated, or formalin-inactivated) were mixed in triplicate wells. Mannose sensitivity of agglutination was determined by adding 100 µL of 5% D-mannose in PBS (Sigma-Aldrich) or PBS only to the wells prior to adding yeast and bacteria. Plates were incubated static at 4°C for 18–24 h. Agglutination was visible macroscopically and images were captured with an iPhone 11 camera. Some samples were removed by gentle pipetting and placed onto a microscope slide. Samples were dried, fixed, and stained with Diff-Quik. Images of agglutinated cells were captured with the 100× oil immersion lens of a Nikon TE300 microscope. To determine agglutination titers, bacteria were diluted twofold serially in PBS. Titer was defined as the highest dilution to yield visible agglutination between bacteria and yeast.

## Hemagglutination assay

Hemagglutination assays as a measure of UPEC P-fimbrial adhesive function were carried out as previously described (48) with modifications. Human Type O red blood cells (RBC) (Zenbio, Cary, NC) were washed three times and resuspended to 1.5% in PBS. UPEC was cultured static in LB for 24 h at 37°C, then diluted 1:100 and cultured for an additional 24 h under the same conditions. Live, PUVA-inactivated, or formalin-inactivated bacteria were washed and adjusted to $OD_{600}$ = 1.0 in PBS. Assays were performed on glass slides by gently mixing 25 µL of bacteria, 25 µL of RBC, and 50 µL of either 2% D-mannose (to measure mannose-resistant hemagglutination, MRHA) or 50 µL PBS. Hemagglutination was observed as clumping of bacteria and RBC after 30–60 min at ambient temperature. Agglutination images were captured by photo-microscopy using the 4× lens of an EVOS M5000 imaging system (Thermo Fisher Scientific).

## UPEC adhesion to human bladder epithelial cells

Assay for UPEC adhesion to human bladder epithelial cells was performed according to a previously published protocol (11) with modifications to facilitate quantitation of cell-associated CFU using live bacteria or cell-associated fluorescence using inactivated bacteria labeled with CFSE. Human bladder epithelial cells, HTB-9 (ATCC 5637), were seeded into a 24-well dish at a density of $1.5 \times 10^5$ cells/well in a volume of 0.5 mL RPMI medium + 10% fetal bovine serum (FBS), L-glutamine, penicillin, and streptomycin. Cells were cultured for 48 h at 37°C + 5% $CO_2$, with 0.5 mL of fresh medium added at 24 h. Static-grown UPEC was inactivated with PUVA or formalin, washed and adjusted to $OD_{600}$ = 0.5 in PBS. Some samples were stained with 5 µM carboxyfluorescein succini-midyl ester (CFSE, BioLegend U.S.) for 30 mi with shaking (80 rpm) at 37°C. Unstained or CFSE-stained UPEC were washed four times with PBS and resuspended in RPMI + 2% FBS to $OD_{600}$ = 0.5. Media were aspirated from HTB-9 monolayers and UPEC were seeded into triplicate wells at an MOI of @20 (500 µL of UPEC, @$2 \times 10^7$ CFU/mL in

RPMI + 2% FBS). Control wells contained HTB-9 cells only or bacteria only. The plate was centrifuged at $600 \times g$ for 5 min and then incubated for 1.5 h at 37°C with 5% $CO_2$. Wells were washed five times with warm Hanks Balanced Salt Solution (HBSS) and once with PBS. For CFU quantitation, 500 µL of 0.1% Triton X100 was added for 10 min to lyse the HTB-9 cells. Each well was gently scraped with the plunger of a 1.0 mL syringe and the lysate was collected into a microfuge tube. Wells were washed with 0.5 mL of PBS and washes were added to tubes for a total volume of 1.0 mL of lysate per sample. Lysates were serially diluted in PBS and plated on LB agar to determine CFU/mL of cell-associated bacteria. For the fluorescence microscopy assay, wells that received CFSE-stained UPEC were washed and fixed with 2% PFA for 30 min at 23°C. HTB-9 cells were then permeabilized with 0.1% Triton X100 in PBS for 5 min. The solution was aspirated and 200 µL of mounting medium [Tris-buffered saline (TBS), pH 8.0 + 2 µg/mL DAPI + 40% glycerol] was added to the wells. Plates were stored sealed and dark at 4°C until analysis. For quantitation of adhesion, the wells were viewed under visible and fluorescent light with the 40× lens of an EVOS M5000 Imaging System equipped with filters for DAPI (HTB-9 cell nuclei) and CFSE (UPEC). The number of CFSE-labeled UPEC and the mean fluorescence intensity (MFI) per 40× field were quantitated using EVOS M5000 Image Analysis Software (version 1.4).

## Assay for hemolytic activity

The hemolysin assay was performed as previously described (49). UPEC was inactivated with PUVA and formalin as indicated except cultures were grown to stationary phase in LB media supplemented with 10 mM $CaCl_2$ prior to treatment with inactivating agents. Live, PUVA-inactivated, or formalin-inactivated UPEC were adjusted to $OD_{600} = 0.1$ in LB + 10 mM $CaCl_2$. Bacteria were then mixed in microfuge tubes with 50 µL washed sheep red blood cells (SRBC #7209003, Lampire Biological Laboratories, Pipersville, PA) in a total volume of 1.0 mL LB + 10 mM $CaCl_2$. For dose-dependence, increasing volumes of bacteria adjusted to $OD_{600} = 0.1$ were mixed with 50 µL SRBC in a final 1.0 mL volume of LB + 10 mM $CaCl_2$. Samples were incubated static at 37°C for 2 or 3 h as indicated with gentle mixing every 20 min. To quantitate hemolytic activity, the cells were pelleted at 3,000 rpm for 5 min in a microfuge, 100 µL of supernatant from each sample was added to a 96-well flat-bottom microtiter plate, and the absorbance at 540 nm was measured using a BioTek Epoch 2 microplate spectrophotometer (Agilent Technologies, Winooski, VT). The extent of RBC lysis was indicated by increased absorbance at 540 nm. Controls included SRBC only, bacteria only and SRBC + 1% Triton-X100 (total lysis). Three independent assays were performed with duplicate or triplicate samples/assays.

## Assay for biofilm formation

A microwell assay for biofilm formation was performed according to published protocols (50, 51). UPEC and *Pseudomonas aeruginosa* strain PA025 control were cultured for 18–24 hr to stationary phase in Mueller-Hinton (MHB) or LB broth, respectively. PUVA or formalin inactivation of UPEC was performed on stationary phase cultures. For the biofilm assay, live, stationary phase UPEC or *P. aeruginosa* were diluted 1:100 into fresh broth and 100 µL was added to quadruplicate wells of a round bottom polystyrene 96-well plate (Nunc). To normalize for the inability of inactivated UPEC to replicate over the assay period, the density of PUVA and formalin-inactivated UPEC was adjusted at the initiation of the assay to the OD reached by live UPEC over a 24-h period ($OD_{600}$ = 4.0) in the microwell cultures. After addition of 100 µL of inactivated bacteria and media controls to quadruplicate wells, the plate was sealed with adhesive and incubated static at 37°C for 24 h. The wells were then emptied, washed three times with distilled water, and biofilms were stained with 125 µL of 0.1% crystal violet solution. After 15 min of staining, the wells were washed four times with water and dried. The crystal violet stain was solubilized for 15 min with 125 µL of 30% acetic acid. To quantitate biofilm formation, 100 µL was transferred to a 96-well flat bottom plate and the absorbance

was measured at 550 nm using a BioTek Epoch 2 microplate spectrophotometer (Agilent Technologies, Winooski, VT).

## Statistics

Experiments were performed at least three times. GraphPad Prism 9.0 software was used to calculate statistical significance between experimental groups by unpaired Student's $t$ test, with $P$ values ≤0.05 considered significant.

## RESULTS

### Inactivation of UPEC with PUVA and formalin

We chose the psoralen drug AMT to photochemically inactivate UPEC according to a protocol that was used to inactivate ETEC (44). An AMT dose-response experiment was performed with doses of 0–100 µg/mL AMT (0–340 µM), followed by irradiation with 2 J/cm² UVA light. AMT doses ≥5 µg/mL yielded no residual live bacteria as determined by plating the entire 1.0 mL culture on LB agar (Fig. 1A). UVA alone or AMT alone did not disable UPEC replication. Controls included heat (1 h 65°C) and 1% formalin. A formalin inactivation dose response is shown in Fig. 1B. A dose of 1.5% was chosen for the formalin-killed UPEC vaccine based on a previous ETEC vaccine study (35). Given a minimal dose of 5 µg/mL AMT to disable UPEC replication, we chose a 10-fold higher dose of 50 µg/mL for subsequent experiments to ensure complete and reproducible inactivation for vaccine purposes. Irreversible inactivation was confirmed at that dose by treating UPEC in triplicate 1.0 mL cultures, washing, and inoculating into 25 mL of fresh LB media. After 72 h at 37°C, there was no change in $OD_{600}$ (Table 1) demonstrating the complete loss of UPEC replication capacity.

### PUVA generates UPEC that are killed but metabolically active

It was previously reported that psoralen-killed bacteria were KBMA due to sustained transcriptional activity in areas of the genome devoid of crosslinks (41). MTS assay was used as a measure of live cell metabolism as reported previously for other species of photochemically inactivated bacteria (37, 42, 52). PUVA-inactivated UPEC was metabolically active after washing and re-inoculating into fresh media, which was not a property of formalin and heat-killed UPEC (Fig. 2A and B). The MTS assay was performed over

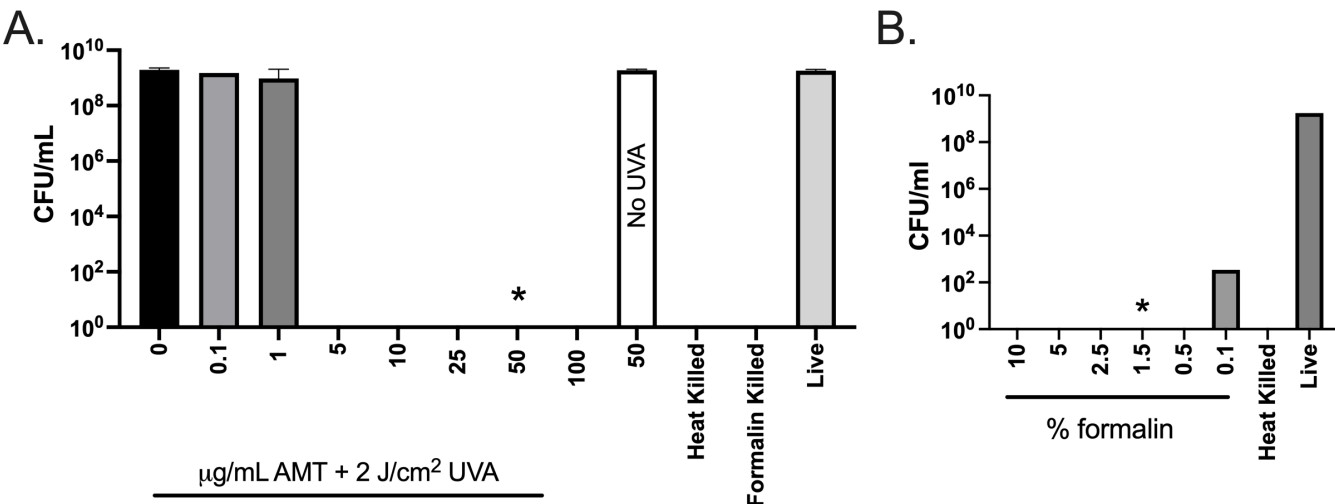

FIG 1 Inactivation of UPEC with PUVA and formalin. (A) UPEC cultures were treated with AMT at the indicated doses (µg/mL) followed by irradiation with UVA light (2 J/cm²), or were treated with heat (1 h, 65°C) or formalin (2 h 37°C, 1.0%). Killing was measured by plating dilutions of treated and untreated cultures on LB agar to quantitate CFU/mL remaining (mean CFU/mL ± S.D., $n$ = 3 per treatment/dose). UVA only, black bar; 50 µg/mL AMT without UVA, white bar; live, untreated bacteria, gray bar. The star indicates the dose of AMT chosen for further study. (B) Formalin treatment (2 h 37°C) over a range of doses, CFU/mL remaining, $n$ = 3. The star indicates the dose of formalin chosen for further study.

**TABLE 1** Complete inactivation of UPEC by PUVA[a]

| Culture | Starting OD$_{600}$ | Final OD$_{600}$ |
|---|---|---|
| PUVA | 0.152 | 0.101 |
| PUVA | 0.140 | 0.098 |
| PUVA | 0.129 | 0.089 |
| Heat killed | 0.106 | 0.107 |
| Live | 0.148 | 3.98 |

[a]OD$_{600}$ of triplicate cultures inactivated with 50 µg/mL AMT and 2 J/cm$^2$ UVA. Starting OD and final OD for each flask, measured at 0 and 72 h, respectively.

an AMT dose range followed by UVA irradiation at 2 J/cm$^2$ (Fig. 2C). Enzymatic activity was detected at all AMT doses, with activity decreasing with increasing dose. A time course analysis was performed to determine if the KBMA state persisted as previously reported (42) (Fig. 2D). Live and PUVA-treated UPEC were washed and inoculated into fresh 96-well LB cultures (1:4) and MTS reagent was added for 2 h at $T = 0$, 2, and 24 h. The results demonstrated enzymatic activity produced by PUVA-inactivated UPEC for up to 24 h. The assay was also performed with automated OD readings over 24 h (Fig. 2E). Although the level and rate of dehydrogenase enzyme activity produced by live and replication-incompetent PUVA-UPEC were different, the absorbance increased steadily for the latter, supporting the conclusion that PUVA-UPEC retained metabolic activity for at least 24 h.

## Immunogenicity of PUVA-UPEC and formalin-UPEC vaccines

We next compared the relative immunogenicity of PUVA- and formalin-UPEC by measuring UPEC-specific antibodies after vaccination of BALB/c mice. Mice were primed and boosted by the IM route with $2 \times 10^7$ CFU equivalents of PUVA-UPEC or formalin-UPEC. Two weeks after the boost, blood was collected and sera were assayed for UPEC-specific IgG by ELISA using whole UPEC as the coating antigen (Fig. 3A and B). The results indicate that the PUVA-UPEC vaccine was highly immunogenic, inducing UPEC-specific serum IgG after prime and boost. The antigen-specific IgG response after vaccination with the PUVA-UPEC vaccine was comparable to that induced by the formalin-UPEC vaccine (geometric mean titers approximately $10^5$ for both vaccines). Therefore, by criteria of the polyclonal serum IgG response to a full repertoire of UPEC antigens presented by a whole bacteria, PUVA-UPEC was as immunogenic as a formalin-killed counterpart.

## Integrity of surface fimbrial proteins measured by yeast agglutination assay

Having determined that PUVA-UPEC is highly immunogenic, we sought to compare the properties of the two vaccines. As a test of the hypothesis that PUVA inactivation would spare proteins from modification, we addressed the integrity of surface adhesins, which are targets of protective immune responses. The rationale was that retention of adhesive function would indicate that surface fimbrial proteins remained structurally intact, and by extension antigenically intact, in PUVA-inactivated bacteria. A yeast agglutination assay that measures mannose-sensitive adhesion of UPEC to yeast cells mediated by T1F (12, 53) was utilized as a read-out of T1F integrity. To maximize T1F production, UPEC were grown under static conditions (47). ELISA using rabbit anti-FimA antibody demonstrated that T1F were present on statically-grown UPEC (Fig. 4A). To measure mannose-sensitive adhesion mediated by T1F, UPEC was washed and incubated with yeast in the presence and absence of 5% D-mannose for 1 hr. Bright-field microscopy demonstrated the adhesion of bacteria to yeast that was inhibited by excess mannose (Fig. 4B). Next, twofold dilutions of live UPEC, PUVA-UPEC and formalin-UPEC were mixed with yeast in a 96-well round bottom plate and agglutination reactions were observed visually. An example is shown in Fig. 4C. Agglutination titers from three independent experiments are shown in Fig. 4D. While in two of the three experiments, the PUVA agglutination titer

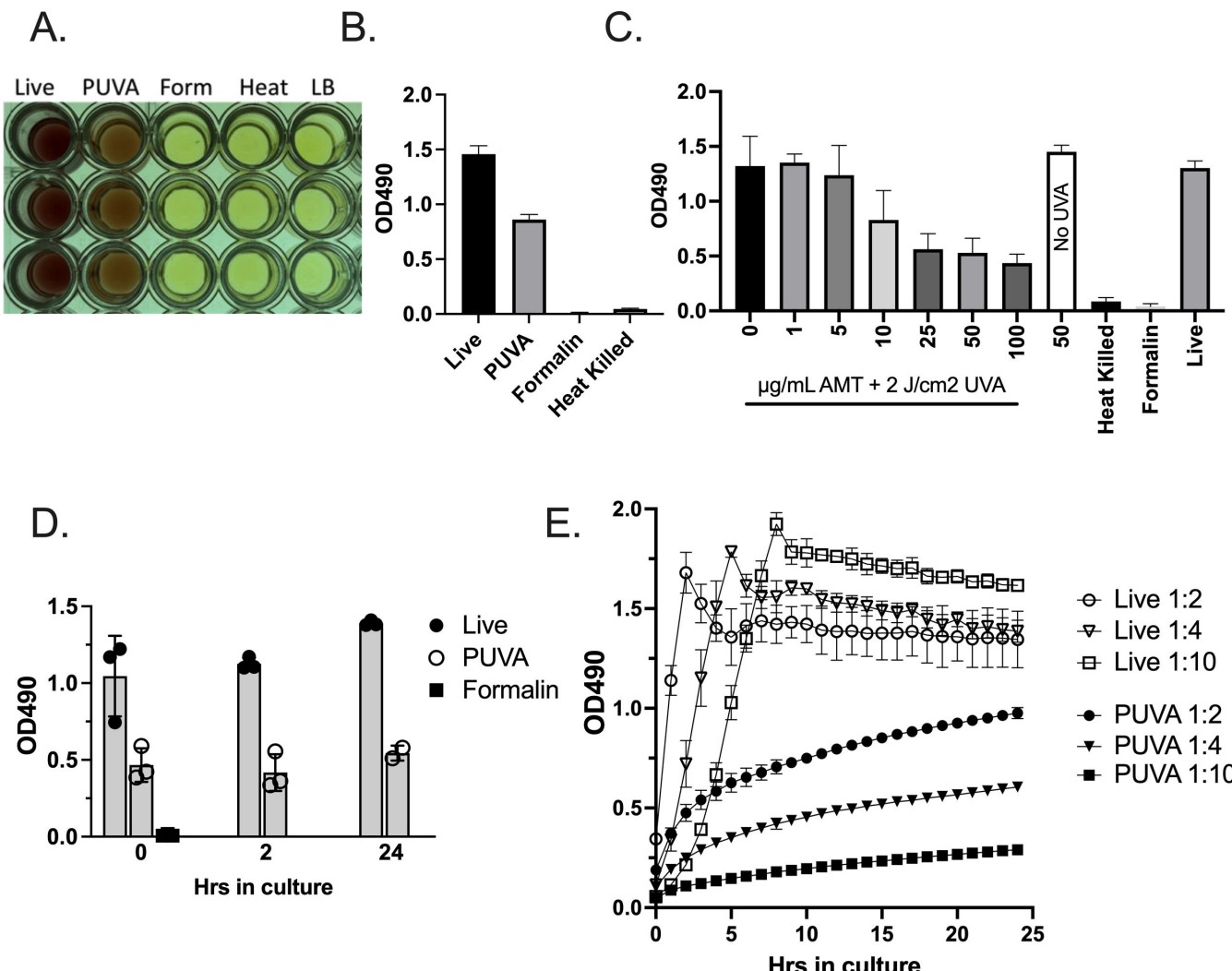

FIG 2   MTS assay for metabolic activity. Live or inactivated UPEC was washed and diluted 1:4 into fresh LB medium + MTS reagent. After a 2-h incubation at 37°C, the absorbance was measured at 490 nm. (A) Photograph of 96-well plate: Live (untreated UPEC); PUVA (UPEC treated with 50 µg/mL AMT + 2 J/cm$^2$ UVA); Form (UPEC treated for 2 h with 1.5% formalin); Heat (UPEC treated for 1 h 65°C); LB (media only). (B) Mean OD$_{490}$ nm ± S.D., triplicate determinations from cultures inactivated under conditions described in panel (A). (C) Metabolic activity, depicting triplicate cultures treated with the indicated doses of AMT + 2 J/cm$^2$ UVA. (D) Metabolic activity measured over time in untreated (live), PUVA-inactivated (50 µg/mL AMT + 2 J/cm$^2$ UVA) or formalin-inactivated (1.5%) cultures; MTS reagent was added for 2 h at the culture time points indicated on the X axis. (E) MTS was added to live or PUVA-treated cultures (50 µg/mL AMT + 2 J/cm$^2$) at time 0 and automated readings were taken every hour at 37°C using a BioTek plate reader with temperature adjustment capabilities. Dilutions of bacteria into fresh media at culture initiation are indicated in the legend. Mean ± S.D. of triplicate values is shown. Panels (B–E) depict MTS activity from independent inactivation experiments.

was reduced twofold compared to live bacteria, formalin inactivation was more detrimental to adhesive function in this assay (live vs formalin and PUVA vs formalin, $P < 0.05$). We conclude that the functional integrity of T1F proteins was retained after PUVA treatment but was negatively impacted by formalin treatment.

## Integrity of surface adhesion proteins by hemagglutination assay

As an additional test of the functional integrity of surface adhesins, a hemagglutination (HA) assay was performed (Fig. 5). This assay detects the presence of P-fimbriae by mannose-resistant binding of the fimbrial tip adhesion, PapG (54) to α-D-galactopyrano-syl-(1-4) β-D-galactopyranoside residues on human type O red blood cells (55). Live or inactivated bacteria and RBC diluted in PBS + mannose were mixed on a microscope

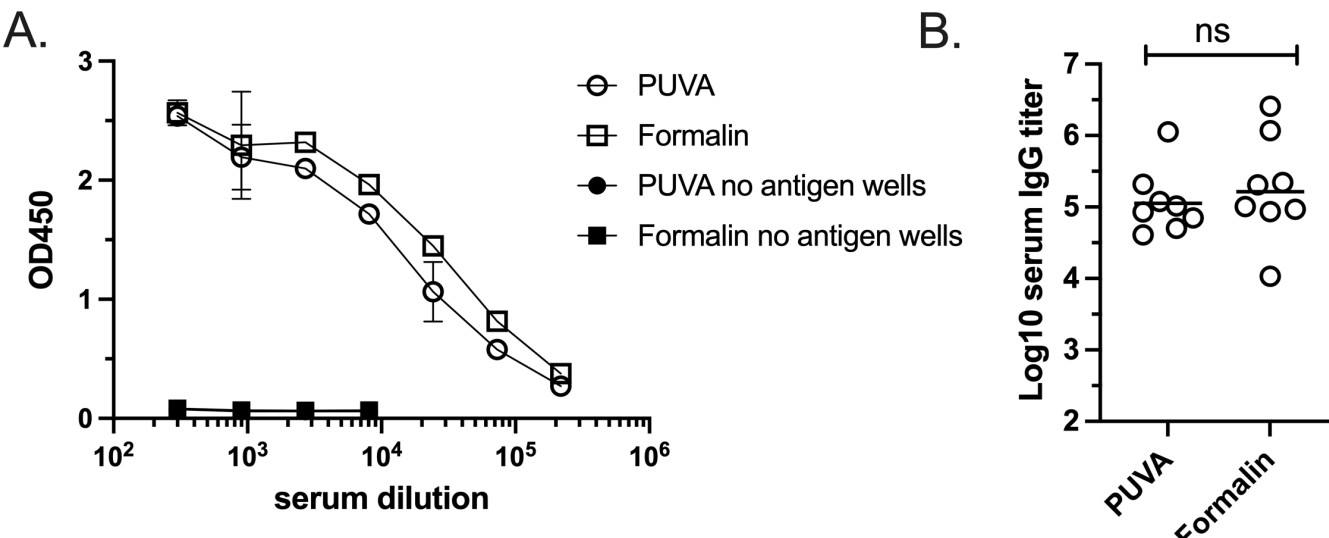

**FIG 3** Immunogenicity of PUVA-UPEC and formalin-UPEC in mice. Mice were vaccinated with PUVA- or formalin-inactivated UPEC and boosted at 3 weeks. Serum IgG reactive with UPEC as the ELISA coating antigen was measured 14 days after boosting. (A) $OD_{450}$ values obtained with pooled sera from the two vaccine groups (open symbols); $OD_{450}$ values of control wells with no antigen coating (closed symbols). The mean ± S.D. of triplicate values is shown. (B) Geometric mean IgG titers from sera of individual mice, which were not significantly different (ns) ($P > 0.05$) as determined by unpaired Student's $t$ test.

slide and photographed at 1 h. Both live and PUVA-inactivated UPEC agglutinated RBCs as shown by distinct clumps, while formalin-UPEC had minimal to no activity, appearing similar to the RBC negative control. Therefore, formalin but not PUVA damaged PapG-mediated adhesive function.

### Assay for adhesion of UPEC to human bladder epithelial cells

To assay for adhesion protein integrity in a physiologically relevant system, binding of live, PUVA- and formalin-treated UPEC to human bladder epithelial cell line HTB-9 was measured. First, the ability of live UPEC to adhere was determined by measuring cell-associated CFU (Fig. 6A) after a 1.5-h incubation with HTB-9 cells. Approximately 1.2% of live UPEC was cell-associated, as per a published study (11). Because PUVA-UPEC does not replicate, a fluorescence-based assay for adhesion using CFSE-labeled UPEC was performed. Flow cytometry demonstrated that 85% of UPEC was brightly labeled with CFSE (Fig. 6B). Next, UPEC was inactivated with PUVA and formalin and then CFSE-labeled. Bacteria were incubated with HTB-9 monolayers and 1.5 h later monolayers were processed for microscopy. The number of CFSE-labeled bacteria and mean fluorescence intensity per 40× field were analyzed using an EVOS 5000 imaging system. As shown in Fig. 6C, the number of live versus PUVA-inactivated cell-associated UPEC was not significantly different. However, the number of formalin-UPEC associated with HTB-9 cells was significantly reduced. The MFI per 40× field was similar for live and inactivated bacteria, suggesting that the reduction in cell-associated formalin-inactivated bacteria was not due to reduced CFSE labeling (Fig. 6C). The experiment was repeated with the addition of excess mannose (Fig. 6D). In each case mannose reduced the binding of UPEC to HTB-9, consistent with binding mediated by T1F. These results indicate that the integrity of UPEC surface adhesins that promoted binding to human bladder cells (presumably including but not necessarily restricted to T1F) was preserved after PUVA treatment but was compromised by formalin.

### Virulence properties of live and inactivated bacteria

Given the KBMA status of PUVA-UPEC, we next determined if the bacteria retained properties associated with virulence that could impact vaccine safety. First we determined if PUVA-UPEC produced hemolytic activity. During infection, an α-hemolysin

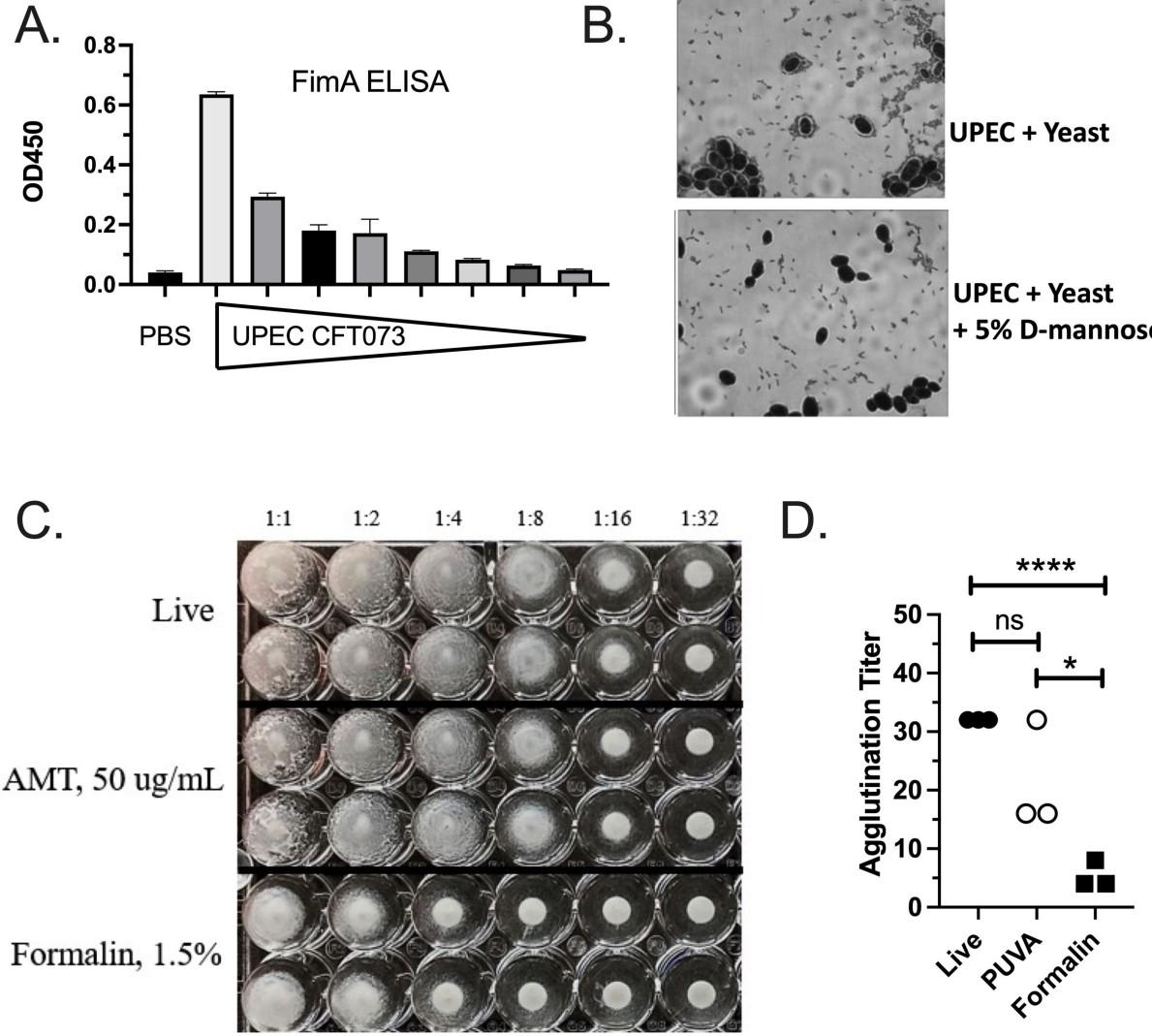

**FIG 4** Yeast agglutination mediated by Type I fimbriae. (A) ELISA to detect T1F production; UPEC grown under static conditions was used as the ELISA coating antigen; tick marks on the *x*-axis represent threefold dilutions of rabbit anti-FimA antibody starting at 1:1,000. PBS indicates the no antigen control wells. The mean $OD_{450}$ value ± S.D. of triplicate wells from two independent experiments is shown. (B) Photomicrographs of UPEC mixed with yeast ±5% D-mannose. Images were captured with a Nikon Eclipse TE300 microscope using a 100× oil immersion lens. (C) UPEC was diluted twofold in round-bottom 96 wells as indicated and mixed with an equal volume of yeast. Agglutination reactions were photographed 24 hr later. One of the three experiments with similar results is shown. (D) The agglutination titer from three experiments, defined as the highest dilution of bacteria at which agglutination was visible is shown in the graph. ****$P < 0.0001$; *$P = 0.04$, unpaired Student's *t* test.

causes damage to the bladder mucosa and promotes persistence of UPEC within superficial bladder epithelial cells (18, 56). Live UPEC was mixed with Sheep RBC (SRBC) in LB broth supplemented with $CaCl_2$. Supernatants were collected 3 h later and the OD at 540 nm was measured as a read-out of SRBC lysis. Live UPEC produced hemolytic activity, which increased with bacterial dose (Fig. 7A). Control samples (SRBC only and bacteria only) did not yield an increase in OD. Next, live UPEC, PUVA-UPEC, formalin-UPEC, and a nonpathogenic strain of *E. coli* (BW25993, CGSC strain 7639) were tested. Live UPEC produced hemolytic activity in a dose-dependent manner, but no activity was produced by PUVA-UPEC, formalin-UPEC, or the *E. coli* control (Fig. 7B). Plating for CFU confirmed that no residual live bacteria remained in the inactivated samples and MTS assay performed in parallel with the hemolysin assay confirmed that PUVA-UPEC was KBMA; mean $OD_{490}$ ± S.D.: live UPEC, 1.57 ± 0.09; PUVA-UPEC, 0.45 ± 0.09; and formalin-UPEC, 0.08 ± 0.01. The capacity to form biofilm-like structures protects UPEC from

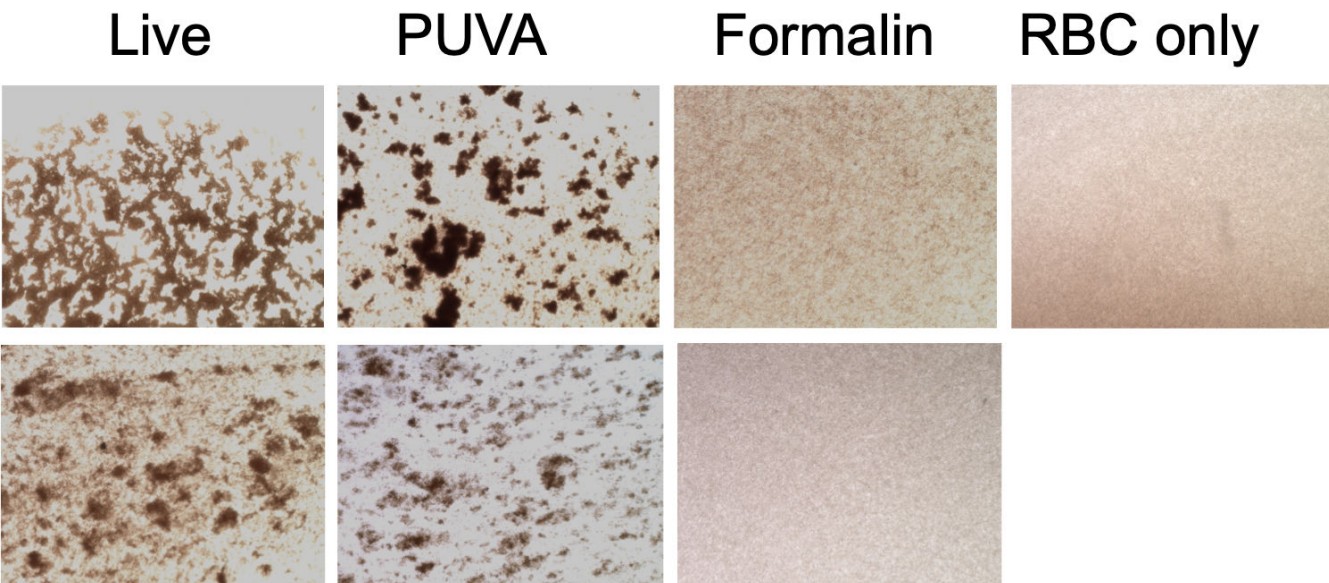

**FIG 5** UPEC and RBC were gently mixed on glass slides and allowed to settle for 1 h at 23°C. Photomicrographs were captured with an EVOS M5000 imaging system using the 4× lens. Two fields for each bacterial preparation are shown along with one field depicting RBC-only control.

immune mediators and antibiotics (20, 21). We tested the capacity for PUVA-UPEC to form biofilms using a microwell assay (50) (Fig. 7C). *P. aeruginosa,* included as a positive control, produced biofilms as did live UPEC CFT073. However, PUVA-UPEC was not capable of forming biofilms, nor as expected were formalin-UPEC, providing additional evidence that KBMA UPEC did not retain key virulence properties of live bacteria.

## DISCUSSION

Vaccines consisting of heat or formalin-killed uropathogenic bacteria or cell lysates have shown various levels of protection against recurrent UTI in animal models and human clinical trials, but improvements that enhance immunogenicity, efficacy, and durability of protection are needed (57). Here, we addressed the potential for improvement of whole-cell UTI vaccines by addressing the inactivation method. Heat and formalin damage proteins, as does ß-propiolactone (BPL) which has been reported for influenza virus vaccines (58). Photochemical inactivation with PUVA crosslinks nucleic acid but spares proteins from modification (32, 38, 40). PUVA inactivation of Dengue virus produced a candidate vaccine with enhanced immunogenicity in mice and NHP relative to a formalin-killed counterpart (59). We have revisited the PUVA approach for bacterial vaccines [reviewed in reference (41)] with a focus on pathogenic *E. coli*, for which safe and effective vaccines are still needed. In this study, we evaluated photochemical inactivation to produce a vaccine for UPEC. We determined the parameters to inactivate UPEC with AMT psoralen and UVA light and examined the properties and immunogenicity of PUVA-inactivated UPEC in comparison to formalin-inactivated UPEC. Both vaccines were highly immunogenic in mice after intramuscular prime and boost, but the PUVA-inactivated vaccine retained properties of live bacteria that have the potential to enhance vaccine performance.

In mice, IM prime/boost with $2 \times 10^7$ CFU equivalents of PUVA- and formalin-inactivated UPEC vaccines gave rise to similar high titer UPEC-specific serum IgG responses. However, when the properties of the inactivated bacteria were examined using functional adhesion as a proxy for protein conformational integrity, we found that formalin negatively impacted proteins. PUVA-UPEC retained full functional adhesive capacity mediated by Type I and P fimbriae as determined by *in vitro* assays including binding to human bladder epithelial cells, while formalin-UPEC was severely defective for adhesion. Considering the role of both types of fimbriae in UPEC pathogenesis and

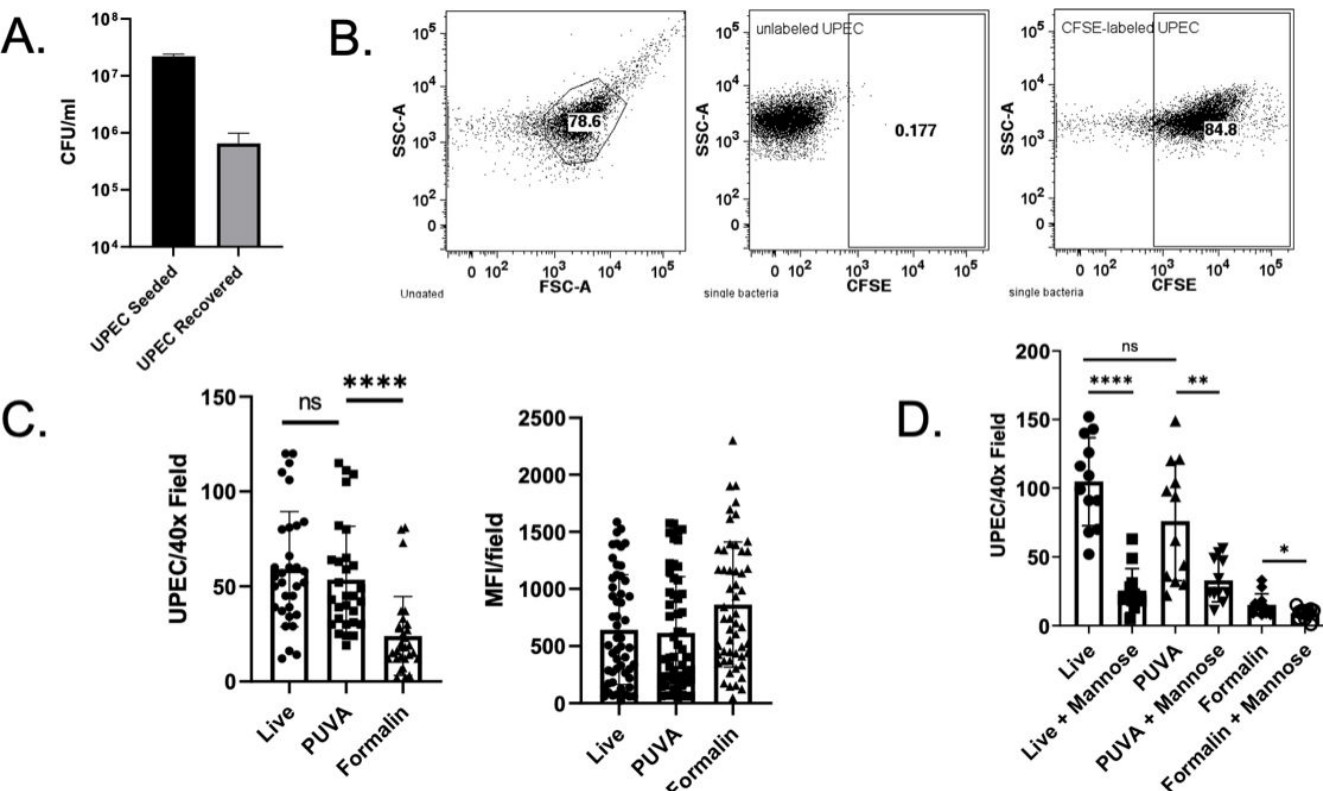

**FIG 6** Adhesion of live and inactivated UPEC to HTB-9 cells. (A) Binding of live UPEC to HTB-9 cells as measured by CFU recovered from washed monolayers after 1.5 h incubation. (B) Flow cytometry showing CFSE labeling of UPEC. (C) Left panel: CFSE-labeled UPEC bound to HTB-9 monolayers. Points represent the number of CFSE-labeled bacteria per 40× field. The mean bacterial count ±S.D. of five wells per sample type, six fields per well, from two independent experiments is shown. Right panel: MFI of multiple fields per sample. The mean MFI ± S.D. is shown by the bar. Statistical significance was determined by unpaired Student's *t* test; ****$P \leq 0.0001$. (D) UPEC binding to HTB-9 cells in the presence and absence of mannose. The mean bacterial count ±S.D. from three wells per sample type, four fields per well, from one experiment is shown. A replicate experiment yielded similar results. Statistical significance was determined by unpaired Student's *t* test; *$P \leq 0.05$, **$P \leq 0.01$, ****$P \leq 0.0001$.

host response in the urinary tract this finding is significant. Type 1 fimbriae mediate adhesion and invasion of UPEC into bladder epithelial cells (11, 60) and antibodies generated against the FimH tip adhesin protect against bladder infection in animal models (13, 61). P-fimbriae mediate adhesion to glycolipid receptors on epithelial cells of the upper urinary tract and antibodies that recognize the PapG tip adhesin were protective in a nonhuman primate (NHP) model of experimental pyelonephritis (62). Consistent with our findings, it was reported that a formalin-inactivated P-fimbriated *E. coli* vaccine lost the ability to bind to P-fimbrial receptors *in vitro*. Using an NHP vaccination model in that study, it was concluded that the production of functional anti-P-fimbrial antibodies capable of blocking adhesion of the homologous challenge strain was compromised by formalin (63). In that context, preservation of fimbrial and other protein antigens presented by a psoralen-inactivated vaccine could be predicted to enhance the production of functional antibodies. In our study, there was no significant difference in the polyclonal anti-UPEC IgG response driven by the two vaccines. However, sera were screened against the full repertoire of protein and polysaccharide antigens presented by whole UPEC, which would not have revealed differences in IgG levels produced against individual proteins. These results are consistent with our recently reported results in which ETEC-specific polyclonal anti-ETEC IgG levels in the serum were not different after vaccination of mice with PUVA-ETEC and formalin-ETEC, but IgG specific for several conserved, outer membrane proteins was significantly greater after vaccination with PUVA-ETEC (44). A similar screen against UPEC proteins that give rise

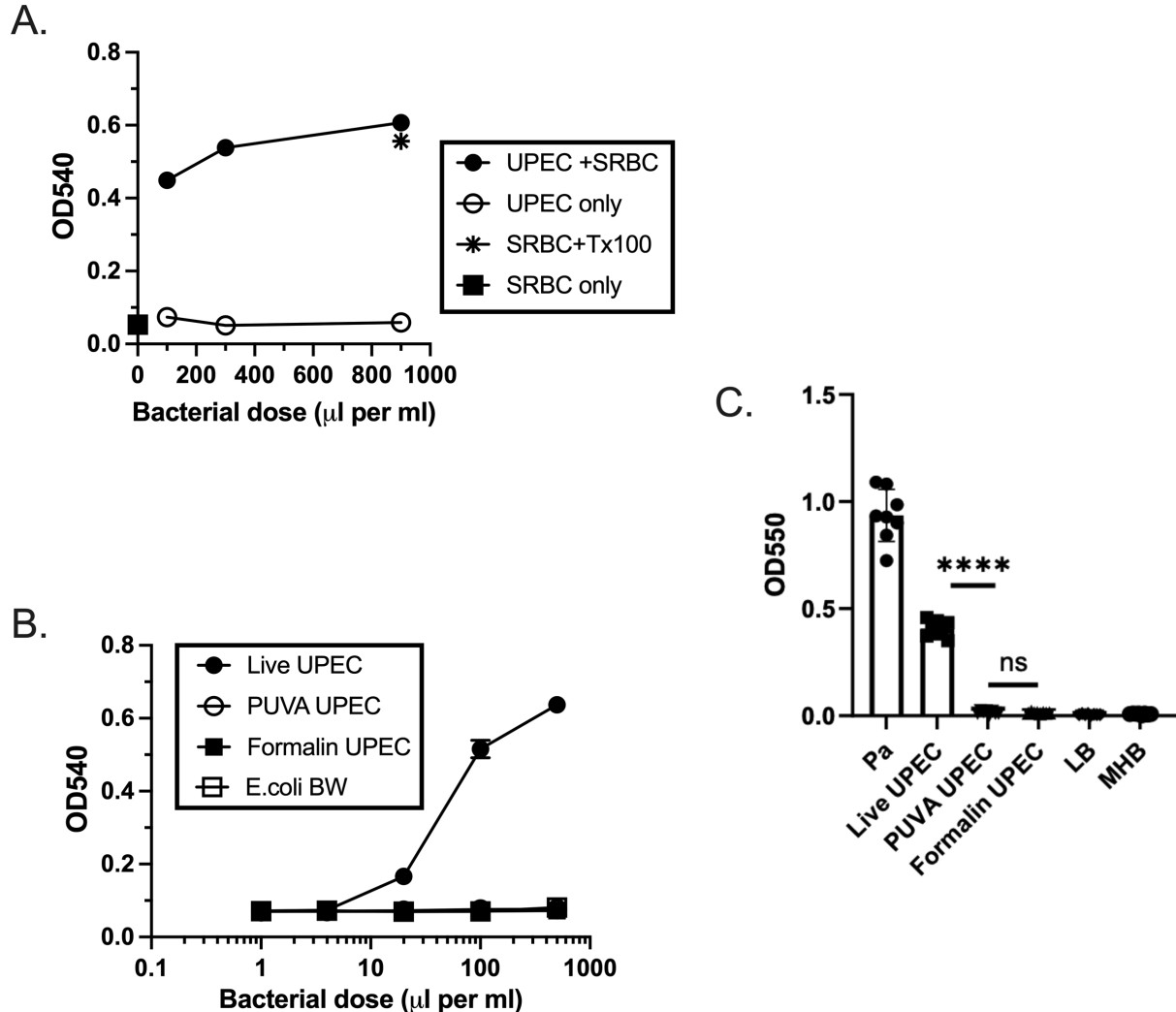

**FIG 7** Hemolytic activity and biofilm production, live and inactivated UPEC. (A) Hemolytic activity produced by live UPEC after 3 h of incubation with SRBC at 37°C; increasing volumes of UPEC (adjusted to OD 0.1) were added to a total volume of 1.0 mL LB + 10 mM $CaCl_2$ + 5% SRBC; SRBC + Tx100 represents total SRBC lysis; (B) Hemolytic activity after 2 h incubation of increasing volumes of live UPEC, PUVA-UPEC, formalin-UPEC, or *E. coli* strain BW25993 with SRBC; mean ± S.D. of triplicate samples is shown. (C) Biofilm formation; microwells were inoculated with 100 µL of stationary phase bacteria: Pa, *P. aeruginosa*, diluted 1:100 into LB broth; Live UPEC, diluted 1:100 into Mueller-Hinton broth (MHB); PUVA-UPEC, adjusted to $OD_{600}$ 4.0 in MHB, formalin-UPEC, adjusted to $OD_{600}$ 4.0 in MHB; LB only or MHB only. After 24 h of static culture at 37°C, biofilms were stained with crystal violet and quantified by measuring the absorbance at 550 nm.

to protective immune responses such as fimbrial and iron-binding proteins and proteins that are conserved across strains (64) would provide key information on the potential of psoralen inactivation to improve whole-cell UPEC vaccines.

As demonstrated previously for other bacterial species (37, 42, 43, 52, 65) and our recent study (44), UPEC was KBMA after PUVA treatment as measured by sustained activity of cellular dehydrogenase enzymes. Previous reports of the KBMA property of psoralen-killed bacteria [reviewed in reference (41)] focused on strains with mutations in the *uvrABC* locus, which renders bacteria highly sensitive to UV damage by disabling the nucleotide excision repair (NER) system (66). The use of *uvr* mutants reduced the frequency of psoralen-induced crosslinks required to disable replication, which allowed for maximal production of endogenous or engineered protein antigens. We based PUVA inactivation of UPEC on our work with ETEC (44) in which wild-type bacteria were rendered replication-defective using the psoralen drug AMT. AMT is one of a group of psoralen drugs (39) that had previously been used to inactivate viruses (32, 38) but not

bacteria. Since our primary goal was to completely inactivate replication while sparing proteins from chemical modification of antigens, which has been documented to occur with formalin-killed vaccines (31, 32, 34–36), we tested AMT doses that were 10- to 100-fold higher on a molar basis than reported for other psoralen drugs used on bacteria (37, 42, 43, 52, 65). Similar doses of AMT and UVA were effective to inactivate ETEC and UPEC. Because crosslink frequency is proportional to psoralen drug concentration (37) the high dose of AMT used in our studies with wild-type bacteria likely induced damage that exceeded the capacity for UV damage repair. However, given that reversion to replication competence would present a significant safety issue, the use of an *uvr* mutant strain to further explore the potential of psoralen-inactivated UPEC as an alternative platform for UTI vaccines is warranted.

KBMA vaccines transiently maintain the ability to produce and secrete proteins *in vitro* and in animals as measured by the induction of immune responses (37, 42, 52, 65). While that property may promote vaccine immunogenicity and in some cases protection (37, 42, 67), it could also present safety issues for non-attenuated strains such as those reported here. Our previous study demonstrated that PUVA-ETEC gave rise to IgG specific for heat-labile toxin (LT), providing evidence that the vaccine produced toxin *in vivo*, which was not a property of formalin-inactivated ETEC (44). In this study, live UPEC produced hemolytic activity and biofilms *in vitro*, both contributing to pathogenicity (10, 68), but PUVA-UPEC did not retain those functions, indicating that some virulence properties were lost. More work is needed to evaluate the balance between toxicity and immunogenicity of psoralen-inactivated pathogenic *E. coli*; however, application of PUVA to attenuated strains as per studies with *Bacillus anthracis* and others (41, 42) is a straightforward approach to enhance safety.

Delivery of vaccines via mucosal routes, such as oral (29), sublingual (16), intranasal (15), and intravaginal (69), is the focus of research for UPEC vaccines and other pathogens that enter and reside at mucosal surfaces (70). In this study, we administered vaccines to mice intramuscularly to compare immunogenicity. As per our study with PUVA-ETEC (44), we observed no adverse effects from vaccination with $2 \times 10^7$ CFU equivalents of PUVA-UPEC as measured by body condition and weight gain over time. However, since parenteral routes are not practical for Gram-negative bacterial vaccines due to the potential for endotoxin-mediated adverse effects, ongoing work is directed toward evaluating local IgG and IgA responses after delivering PUVA-UPEC directly to the urogenital tract of mice, modeling a translationally relevant route of UPEC vaccine delivery (30).

In summary, PUVA inactivation produces highly immunogenic, replication-disabled UPEC which retains a native-like surface. Our data indicate that unlike formalin, PUVA does not alter UPEC proteins as measured by retention of fimbrial adhesive function, a key target of protective immune responses. We acknowledge that protection studies that include comparison to formalin and heat-killed UPEC vaccines using mouse challenge models and clinically relevant routes of vaccination are crucial as a next step toward evaluating the potential of PUVA-UPEC. There are major challenges that remain if PUVA-UPEC demonstrates protection against urinary tract infection in preclinical models. The feasibility of the approach for scale-up is unknown. The translational suitability of KBMA vaccines with respect to toxicity, the potential for reversion, and the stability of protein antigens remain to be determined. With these challenges acknowledged, the PUVA vaccine platform has potential to improve existing, suboptimally protective UTI vaccines by leveraging the full complement of antigens (polysaccharides and proteins) for presentation to the immune system in a configuration that resembles infectious bacteria. Future studies could test this new approach in mouse (7) and NHP challenge models (71) in combination with strategies that have shown potential to improve UTI vaccines including mucosal delivery (28, 30, 72) and combination with adjuvants that skew immune responses toward protective Th1-type immunity (24).

## ACKNOWLEDGMENTS

This research was funded by a Department of the Navy subcontract award #N626451920002 (J.S.) and a WFU Graduate Student Research Award (A.E.M.). The authors thank Dr Sargaru Subashchandrabose (School of Veterinary Medicine and Biomedical Sciences, Texas A&M) for reagents and protocols.

## AUTHOR AFFILIATIONS

[1]Department of Microbiology and Immunology, Wake Forest University School of Medicine, Winston Salem, North Carolina, USA
[2]Department of Internal Medicine, Infectious Diseases Section, Wake Forest University School of Medicine, Winston Salem, North Carolina, USA

## AUTHOR ORCIDs

Marlena M. Westcott http://orcid.org/0000-0002-6832-7355

## FUNDING

| Funder | Grant(s) | Author(s) |
| --- | --- | --- |
| U.S. Department of Defense (DOD) | N626451920002 | John W. Sanders |

## AUTHOR CONTRIBUTIONS

Marlena M. Westcott, Conceptualization, Formal analysis, Investigation, Methodology, Project administration, Supervision, Validation, Visualization, Writing – original draft, Writing – review and editing | Alexis E. Morse, Formal analysis, Funding acquisition, Investigation, Methodology, Visualization, Writing – review and editing | Gavin Troy, Investigation | Maria Blevins, Investigation | Thomas Wierzba, Investigation | John W. Sanders, Conceptualization, Funding acquisition, Project administration, Resources, Supervision, Validation, Writing – review and editing

## ADDITIONAL FILES

The following material is available online.

Open Peer Review

**PEER REVIEW HISTORY (review-history.pdf).** An accounting of the reviewer comments and feedback.

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
