## [Reviewer comments · Microbiology Spectrum]

Microbiology Spectrum

Photochemical inactivation as an alternative method to produce a whole cell vaccine for uropathogenic *E. coli* (UPEC)

Marlena Westcott, Alexis Morse, Gavin Troy, Maria Blevins, Thomas Wierzba, and John Sanders

Corresponding Author(s): Marlena Westcott, Wake Forest University School of Medicine

Review Timeline:

Submission Date:	October 20, 2023
Editorial Decision:	December 3, 2023
Revision Received:	January 2, 2024
Accepted:	January 10, 2024

Editor: Shengqing Yu

Reviewer(s): Disclosure of reviewer identity is with reference to reviewer comments included in decision letter(s). The following individuals involved in review of your submission have agreed to reveal their identity: Weihui Li (Reviewer #2)

Transaction Report:

DOI: <https://doi.org/10.1128/spectrum.03661-23>

Re: Spectrum03661-23 (Photochemical inactivation as an alternative method to produce a whole cell vaccine for uropathogenic E. coli (UPEC))

Dear Dr. Marlena M Westcott:

Thank you for the privilege of reviewing your work. Below you will find my comments, instructions from the Spectrum editorial office, and the reviewer comments.

Revision Guidelines

Sincerely,
Shengqing Yu
Editor
Microbiology Spectrum

Reviewer #1 (Comments for the Author):

Westcott and colleagues have performed experiments aimed at developing a vaccine based on a photochemically inactivated E. coli to protect against urinary tract infection. This manuscript is a follow-up study to reference #44 and contains important observations contextualizing the prior work. The methods section is extremely thorough and appears to include every detail. Experiments include all necessary controls. Results clearly show that the photochemically inactivated E. coli are more similar to

live bacteria than formalin-fixed bacteria are, i.e. photochemically inactivated bacteria "retained properties of live bacteria that have potential to enhance vaccine performance" (line 386-387), which explains their immunogenic superiority as reported in reference #44. Appropriate statistical tests have been included in the analysis of results. The authors appear to have paid meticulous attention to every detail. I have no suggestions for improvements to this manuscript. Great work!

Reviewer #2 (Comments for the Author):

Here are some suggestions:

1. Is the treatment condition for the PUVA group in Figure 2B 50 µg/ml AMT? The same experimental conditions in Figure 2C seem to produce inconsistent results, OD490 approximately 0.8 in Figure 2B and approximately 0.5 in Figure 2C. Maybe a data check in order.
2. Line 702, "One of 3 experiments with similar results is shown." The data from the other two parallel experiments should be listed. Line 321, "Live and PUVA-UPEC had similar agglutination titers (1:16) while the titer for formalin-UPEC was decreased to 1:4." The right side Figure 4C, the agglutination titer of Live (1/32), PUVA (1/16), Formalin (1/8). The data from the other two parallel experiments is crucial for supporting the conclusion.
3. Line 457, "Protection studies that include comparison to formalin and heat-killed UPEC vaccines using mouse challenge models and clinically relevant routes of vaccination are needed as a next step toward evaluating the potential of PUVA-UPEC." Although UPEC-specific ELISA and some other experiments are used to test the performance of vaccines, protective experiments are crucial for evaluating vaccine efficacy.

Response to Reviewers, Westcott, M, *et al.*, Photochemical Inactivation as an alternative method to produce a whole cell vaccine for uropathogenic *E. coli*.

Reviewer 1:

No issues raised

Reviewer 2:

We thank the reviewer for thoughtful comments which we have responded to as follows to improve the manuscript:

1. *Is the treatment condition for the PUVA group in Fig 2B 50 µg/ml AMT?*

Response: Yes. This detail has been added to Fig. 2B legend (lines 670-671).

The same experimental conditions in Fig 2C seem to produce inconsistent results, OD490 @ 0.8 in Fig. 2B and @ 0.5 in Fig. 2C.

Response: The reviewer correctly notes that the OD490 values for PUVA in Fig. 2C are lower than in 2B for the 50 µg/ml AMT dose. The data depicted in each panel, B – E, reflect independent inactivation experiments using 50 µg/ml AMT (B, D, E) or multiple doses of AMT (C). The differences in OD490 reflect variability of dehydrogenase activity levels produced by a bacterial population prepared on a given day, which is also manifested by variability in activity produced by live bacteria controls in the different panels. Detail has been added to the legend to clarify inactivation conditions for each panel and to point out independent inactivation experiments (lines 670-679).

2. *Line 702, “One of 3 experiments with similar results is shown.” The data from the other 2 parallel experiments should be listed; Line 321, “Live and PUVA-UPEC had similar agglutination titers (1:16) while the titer for formalin-UPEC was decreased to 1:4....The data from the other two parallel experiments is crucial for supporting the conclusion.”*

Response: Both comments refer to the data depicted in the bar graph in Fig. 4C. As the reviewer points out, all 3 experiments should have been depicted in support of the conclusion. The figure has been modified to include the agglutination titers from 3 independent experiments. In the revised Fig. 4, the agglutination titer graph is shown in a separate panel D. The Fig. 4 legend has been updated accordingly (lines 710-712) and the Results text has been corrected to reflect data presented from 3 independent experiments (lines 321-326).

3. The reviewer notes that protection experiments are crucial for evaluating vaccine efficacy. The text in lines 461-462 of the Discussion now emphasizes that point.

Re: Spectrum03661-23R1 (Photochemical inactivation as an alternative method to produce a whole cell vaccine for uropathogenic E. coli (UPEC))

Dear Dr. Marlena M Westcott:

Your manuscript has been accepted, and I am forwarding it to the ASM production staff for publication. Your paper will first be checked to make sure all elements meet the technical requirements. ASM staff will contact you if anything needs to be revised before copyediting and production can begin. Otherwise, you will be notified when your proofs are ready to be viewed.

Sincerely,
Shengqing Yu
Editor
Microbiology Spectrum

Reviewer #1 (Comments for the Author):

I had no suggestions for improvement to the original manuscript, but the revisions made in response to reviewer #2 have strengthened the manuscript.

Reviewer #2 (Comments for the Author):

I don't have any suggestions for improving this manuscript.